# Experimental realization of stimulated Raman shortcut-to-adiabatic passage with cold atoms

Yan-Xiong Du[1], Zhen-Tao Liang[1], Yi-Chao Li[2], Xian-Xian Yue[1], Qing-Xian Lv[1], Wei Huang[1], Xi Chen[2], Hui Yan[1] & Shi-Liang Zhu[1,3,4]

Accurate control of a quantum system is a fundamental requirement in many areas of modern science ranging from quantum information processing to high-precision measurements. A significantly important goal in quantum control is preparing a desired state as fast as possible, with sufficiently high fidelity allowed by available resources and experimental constraints. Stimulated Raman adiabatic passage (STIRAP) is a robust way to realize high-fidelity state transfer but it requires a sufficiently long operation time to satisfy the adiabatic criteria. Here we theoretically propose and then experimentally demonstrate a shortcut-to-adiabatic protocol to speed-up the STIRAP. By modifying the shapes of the Raman pulses, we experimentally realize a fast and high-fidelity stimulated Raman shortcut-to-adiabatic passage that is robust against control parameter variations. The all-optical, robust and fast protocol demonstrated here provides an efficient and practical way to control quantum systems.

[1] Guangdong Provincial Key Laboratory of Quantum Engineering and Quantum Materials, SPTE, South China Normal University, Guangzhou 510006, China. [2] Department of Physics, Shanghai University, Shanghai 200444, China. [3] National Laboratory of Solid State Microstructures, School of Physics, Nanjing University, Nanjing 210093, China. [4] Synergetic Innovation Center of Quantum Information and Quantum Physics, University of Science and Technology of China, Hefei 230026, China. Correspondence and requests for materials should be addressed to X.C. (email: xchen@shu.edu.cn) or to H.Y. (email: yanhui@scnu.edu.cn) or to S.-L.Z. (email: slzhu@nju.edu.cn).

Coherent control of the quantum state is an essential task in various areas of physics, such as high-precision measurement[1,2], coherent manipulation of atom and molecular systems[3,4], and quantum information[5,6]. In most applications, the basic requirement of coherent control is to reach a given target state with high fidelity as fast as possible. Many schemes have been developed for this purpose, including the adiabatic passage technique, which drives the system along its eigenstate[7–10]. One of attractive property of this technique is that the resulting evolution is robust against control parameter variations when the adiabatic condition is fully satisfied. However, the adiabatic passage techniques such as the two-level adiabatic passage[10], three-level stimulated Raman adiabatic passage (STIRAP)[11] and their variants are time consuming to realize, which limits their applications in some fast dephasing quantum systems. To overcome this shortcoming, several protocols within the framework of the so-called 'shortcut-to-adiabaticity'[12] have been proposed to speed-up the 'slow' adiabatic passage: for instance, counter-diabatic driving (equivalently, the transitionless quantum algorithm)[13–16]. Very recently, the acceleration of the adiabatic passage has been demonstrated experimentally in two-level systems: an energy-level anticrossing for a Bose–Einstein condensate loaded into an accelerated optical lattice[17] and the electron spin of a single nitrogen-vacancy centre in diamond[18].

The STIRAP based on the two-photon stimulated Raman transition has several advantages. First, lasers can be focused on a single site in an optical lattice or on a single ion in a linear ion trap, which guarantees individual addressability[19–21]. Second, the STIRAP can couple two states that cannot be directly coupled, such as transferring population between two atomic states with the same parity (which cannot be directly coupled via electric dipole transition)[22], or transferring the atomic state to the molecular state[3]. Furthermore, with large single-photon detuning, double coherent adiabatic passages exist[23–25], which guarantees the capacity for state transfer between arbitrary states[25–27]. Interestingly, several theoretical protocols have been proposed to speed-up the STIRAP by adding an additional microwave field in various atom and molecular systems[28–31]. However, the transfer fidelity will depend on the phase differences among the microwave field, the Stokes and pumping laser pulses for the STIRAP, which are difficult to lock. Furthermore, the combination of the microwave field and Raman lasers makes it difficult to feature the individual addressability of the operation. Therefore, speeding up the STIRAP has not yet been experimentally demonstrated.

Motivated by the goal of a robust, fast, addressable, arbitrary state transfer protocol, we propose a feasible scheme to speed-up STIRAP by modifying the shapes of two Raman pulses. We utilize the counter-diabatic driving along with unitary transformation, one of the shortcut techniques to realize adiabatic passages. We then experimentally demonstrate the proposed stimulated Raman shortcut-to-adiabatic passage (STIRSAP) protocol in a large single-photon detuning three-level $\Lambda$ system with a cold atomic ensemble. The passage's robustness against parameter variation is confirmed in our experiments. Fast, robust, individually addressable and arbitrarily transferable between states, the quantum state control protocol demonstrated here is useful for practical applications.

## Results

**STIRAP and STIRSAP protocols.** We consider a cold $^{87}$Rb atom ensemble (see the Methods section) whose internal energy states $|1\rangle$ ($|2\rangle$) and $|3\rangle$ are coupled by pumping pulse $\Omega_P(t)$ (Stokes pulse $\Omega_S(t)$), as shown in Fig. 1a. Two ground states $|F=1, m_F=0\rangle = |1\rangle$; $|F=2, m_F=0\rangle = |2\rangle$ and one excited state $5^2P_{3/2}$ ($=|3\rangle$) are selected as a typical three-level $\Lambda$ system. Under the conditions of rotating-wave approximation and two-photon detuning $\delta = 0$, the interaction Hamiltonian of the system in the basis of $\{|1\rangle, |2\rangle, |3\rangle\}$ is given as

$$H_\Lambda(t) = \frac{\hbar}{2} \begin{pmatrix} 0 & 0 & \Omega_P(t)e^{i\varphi_L} \\ 0 & 0 & \Omega_S(t) \\ \Omega_P(t)e^{-i\varphi_L} & \Omega_S(t) & 2\Delta \end{pmatrix} \quad (1)$$

where $\Delta$ is the single-photon detuning and $\varphi_L$ is the phase difference between Stokes and pumping lasers, and has been locked to a fixed value in our experiment. In the large detuning condition $\Delta \gg \sqrt{\Omega_P^2(t) + \Omega_S^2(t)}$, the three dressed states of the Hamiltonian (1) can be described as $|D\rangle = \cos\theta|1\rangle - \sin\theta \exp(-i\varphi_L)|2\rangle$, $|B_1\rangle \simeq \sin\theta \exp(i\varphi_L)|1\rangle + \cos\theta|2\rangle$, and $|B_2\rangle \simeq |3\rangle$, where mixing angle $\theta = \arctan[\Omega_P(t)/\Omega_S(t)]$ (refs 25,32). In the usual STIRAP protocol, the Stokes and pumping laser pulses are partially overlapping Gaussian shapes[11]. If the adiabatic condition $T \gg T_\pi$ is fulfilled, where $T$ is the operation time and $T_\pi = 2\pi\Delta/(\Omega_P\Omega_S)$, with $\Omega_P$ and $\Omega_S$ being the respective peaks of the pulses $\Omega_P(t)$ and $\Omega_S(t)$, a high-fidelity coherent population transfer from one specific superposition state of $|1\rangle$ and $|2\rangle$ to another can be realized through adiabatic evolution of the dressed states $|D\rangle$ and $|B_1\rangle$. This protocol is the double coherent STIRAP[25] we used in our experiments.

To release the critical requirement $T \gg T_\pi$ but still maintain the high-fidelity, one can adopt the shortcut approach to adiabatic passage[14–16]. Under the large detuning condition, the population in excited state $|3\rangle$ can be adiabatically eliminated. The Hamiltonian (1) can then be reduced into an effective two-level system on the basis $\{|1\rangle, |2\rangle\}$, and the Hamiltonian is given by

$$H_0(t) = -\frac{\hbar}{2} \begin{pmatrix} \Delta_{\text{eff}} & \Omega_{\text{eff}}e^{i\varphi_L} \\ \Omega_{\text{eff}}e^{-i\varphi_L} & -\Delta_{\text{eff}} \end{pmatrix} \quad (2)$$

where the effective detuning $\Delta_{\text{eff}} = [\Omega_P^2(t) - \Omega_S^2(t)]/(4\Delta)$ and the effective Rabi frequency $\Omega_{\text{eff}} = \Omega_P(t)\Omega_S(t)/(2\Delta)$. According to the standard shortcut approach to adiabatic passage, the diabatic transition can be eliminated by adding an appropriate auxiliary counter-diabatic term $H_{\text{cd}}(t)$ defined in the Methods section[12,16]. In our system, this auxiliary term $H_{\text{cd}}(t)$ can be realized by adding a microwave field to couple the levels $\{|1\rangle$ and $|2\rangle\}$ (refs 29,30); however, the aforementioned drawbacks of this method still need to be overcome.

In the Methods section, we describe a feasible approach to realize the shortcut method to adiabatic passage. We find that high-fidelity STIRSAP can be achieved if the shapes of the Raman pulses are replaced by

$$\tilde{\Omega}_P(t) = \sqrt{2\Delta\left(\sqrt{\tilde{\Delta}_{\text{eff}}^2(t) + \tilde{\Omega}_{\text{eff}}^2(t)} + \tilde{\Delta}_{\text{eff}}(t)\right)},$$
$$\tilde{\Omega}_S(t) = \sqrt{2\Delta\left(\sqrt{\tilde{\Delta}_{\text{eff}}^2(t) + \tilde{\Omega}_{\text{eff}}^2(t)} - \tilde{\Delta}_{\text{eff}}(t)\right)}, \quad (3)$$

where $\tilde{\Delta}_{\text{eff}}(t)$ and $\tilde{\Omega}_{\text{eff}}(t)$ are, respectively, the modified effective detuning and Rabi frequency as defined in the Methods section. The modified Raman pulses still satisfy the large detuning condition. With appropriate choices of the parameters $\tilde{\Omega}_P(t)$ and $\tilde{\Omega}_S(t)$, the system is effectively equivalent to that of adding a supplementary counter-diabatic term $H_{\text{cd}}(t)$ (refs 17,33). The system will thus evolve along its eigenstate of the Hamiltonian $H_0(t)$ up to the phase factor for any choice of the protocol parameters, even with very small values of Stokes and pumping fields, and within an arbitrarily short operation time $T$. According to equation (3), given the original Stokes and pumping pulses

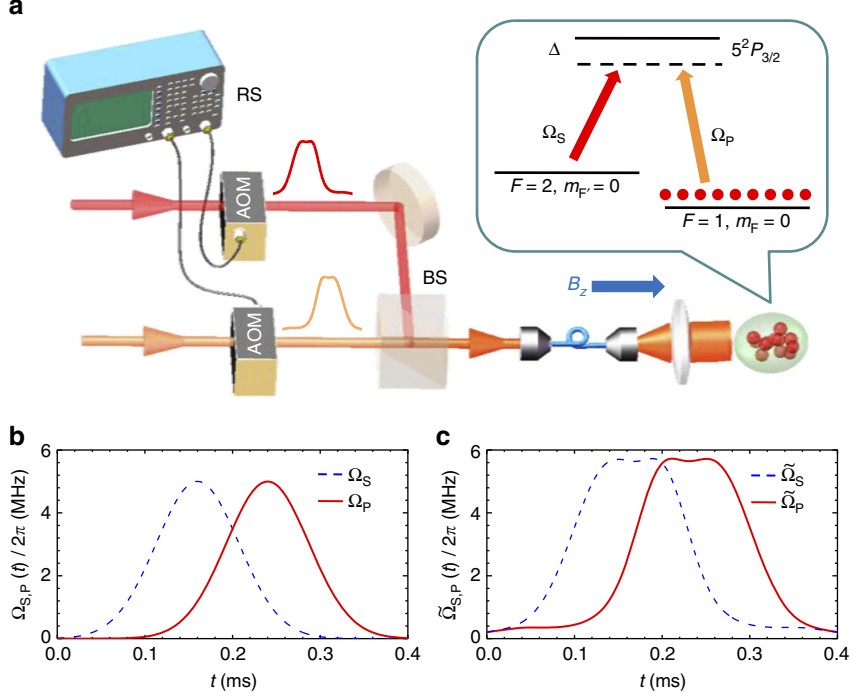

**Figure 1 | Experimental scheme.** (**a**) Experimental set-up. The laser-atom coupling scheme of the three-level $\Lambda$ system is shown in the upper panel. Two ground states $|F=1, m_F=0\rangle = |1\rangle$; and $|F=2, m_F=0\rangle = |2\rangle$, and one excited state $5^2P_{3/2}$ $(=|3\rangle)$ of $^{87}$Rb are selected as a typical three-level $\Lambda$ system. The states $|1\rangle$, $(|2\rangle)$ and $|3\rangle$ are coupled by pumping pulse $\Omega_P(t)$ (Stokes pulse $\Omega_S(t)$). The single-photon detuning $\Delta$ between the Raman lasers and the excited state $5^2P_{3/2}$ is about 2.5 GHz. A magnetic field $B_z$ is used to split the Zeeman sublevels. Two Raman laser fields (pumping $\Omega_P(t)$ and Stokes $\Omega_S(t)$) with phase-locked are combined by a beam splitter (BS) and then send to interact with the cold atoms. The shapes of the Raman lasers are modulated by two AOMs driven by a radio source (RS). (**b**) The original Raman laser pulses in the usual STIRAP are two partially overlapping Gaussian shapes. (**c**) Modified Raman laser pulses for STIRSAP obtained from equation (3).

with the Gaussian-beam shape shown in Fig. 1b, the modified Stokes and pumping pulses required for STIRSAP can be obtained as shown in Fig. 1c.

**Dynamics and characteristics**. We now compare the performance of the above STIRAP and STIRSAP protocols. In our experiment, the Stokes pulse $\Omega_S(t) = \Omega_S \exp\left[-(t-T/2+\Delta\tau)^2/\sigma^2\right]$ and pumping pulse $\Omega_P(t) = \Omega_P \exp\left[-(t-T/2-\Delta\tau)^2/\sigma^2\right]$, where $2\sigma = T/3$ is the full-width at half-maximum of the pulse, and $\Delta\tau = T/10$ is the separation time between the two pulses. We first compare the population transfer dynamics with Raman pulses as shown in Fig. 1b,c. The original parameters of STIRAP are set to be $\Delta \sim 2\pi \times 2.5$ GHz, $\Omega_P = \Omega_S = 2\pi \times 5$ MHz, and hereafter we denote $\Omega_0 \equiv 2\pi \times 5$ MHz and the corresponding $\pi$ pulse time $T_0 \equiv 2\pi\Delta/\Omega_0^2 = 0.1$ ms. Experimental data (blue and red squares) and theoretical results (dashed and solid lines) are shown together in Fig. 2a. Here the operation time $T = 0.4$ ms, which fails to fulfil the adiabatic criteria. As shown in Fig. 2a, the final transfer efficiency of STIRAP only reaches 36% (blue dashed line). As for the STIRSAP Raman pulses implemented by replacing $\Omega_{P,S}(t)$ with $\tilde{\Omega}_{P,S}(t)$ in equation (3), the transfer efficiency (the red solid line) can reach 100% since the diabatic transition has been eliminated by effectively adding the Hamiltonian $H_{cd}(t)$. The peak transfer efficiencies of STIRSAP are observed with a two-photon detuning $\delta = -7$ kHz due to ac-Stark shift. The ac-Stark shift can be viewed as a perturbation in our case since it is small compared with $\Omega_{S,P}$ and the two-photon bandwidth ($\sim 20$ kHz)[25]. The experimental and theoretical results fit very well with each other. This result clearly shows the remarkable feature of the STIRSAP protocol.

To further characterize the performance of STIRAP and STIRSAP, we plot the transfer efficiencies of them as a function of

operation time $T$ in Fig. 2b for a fixed $\Omega_{P,S} = \Omega_0$. With STIRAP, the transfer efficiency approaches 100% when the operation time is longer than $25T_0$, where the adiabatic condition is fully satisfied[11]; however, the efficiency (blue-dashed line) will decrease along with the decreasing of $T$. In particular, it decreases quickly when $T < 10T_0$. Remarkably, it is shown in theoretical calculation that the transfer efficiency of STIRSAP (red solid line) can keep constant for any operation time $T$ since the diabatic transition has been eliminated by effectively adding the $H_{cd}$ term through modifying the shape of the pulses accordingly. We confirm the theoretical result with the experimental data for $T \geq 4T_0$, where the peak of $\tilde{\Omega}_{P,S}(t)$ is around $1.14\Omega_0$ for $T = 4T_0$.

In principle, both STIRAP and STIRSAP can be sped up to a fixed operation time with fidelity higher than certain value if the peaks of Raman pulses are sufficiently large; however, the resources required are different. With STIRAP, we denote the peak of $\Omega_{P,S}(t)$ as $\Omega_{AP}$. Because the characterized time for adiabatic evolution $T_\pi = 2\pi\Delta/\Omega_{AP}^2$ decreases with increasing $\Omega_{AP}$, the operation time can decrease even for a fixed fidelity. By contrast, as shown in Fig. 2b, the operation time for STIRSAP can be arbitrarily small by suitably choosing the peak $\tilde{\Omega}_{SA}$ of the modified Raman pulses $\tilde{\Omega}_{P,S}(t)$. To address the resources required for the speed-ups, we plot in Fig. 2c the peaks $\Omega_{AP}$ (blue dashed line) and $\tilde{\Omega}_{SA}$ (red solid line) required for operation time $T$, with fidelity no less than 99.4%. It is clear that peak $\tilde{\Omega}_{SA}$ is much smaller than $\Omega_{AP}$ for the same operation time with the same high fidelity. This reveals that for the same time $T$ and same fidelity, the resources required for STIRSAP is less than that for STIRAP.

To further compare the performance of STIRAP and STIRSAP, we test the maximum capability of speed-up that we could obtain for equal maximum Rabi frequencies, that is, $\Omega_{AP} = \tilde{\Omega}_{SA}$. We theoretically calculate the time $T_{AP}$ of STIRAP to achieve the

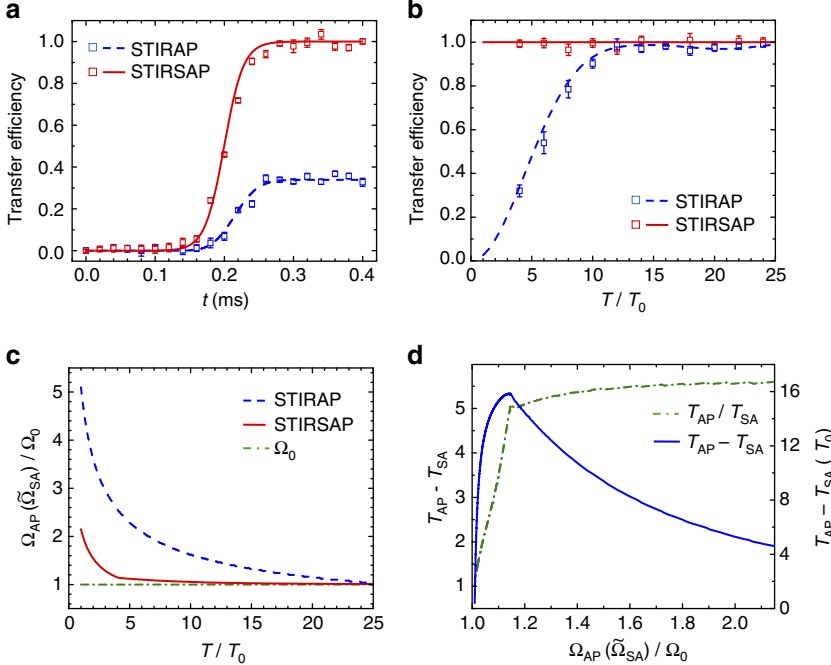

**Figure 2 | The speed-up results.** (**a**) Experimental (squares) and theoretical (lines) results of population transfer dynamics. STIRAP is driven by the Raman pulses plotted in Fig. 1b while STIRSAP is driven by the one in Fig. 1c. The transfer efficiency of STIRAP can only reach 36%; in contrast, that for STIRSAP can approach 100%. (**b**) Transfer efficiency versus operation time $T$. The data points in **a** and **b** are averaged over five measurements, each with the error bars depicting the s.d. (**c**) Maximum Rabi frequency $\tilde{\Omega}_{SA}$ of STIRSAP (red solid line) and $\Omega_{AP}$ of STIRAP (blue dashed line) versus operation time $T$ with the same fidelity. The original Raman frequency $\Omega_0$ is plotted as green dotted–dashed line. (**d**) Comparison of time $T_{AP}$ of STIRAP and $T_{SA}$ of STIRSAP to achieve the same 99.4% efficiency and with equal maximum Rabi frequency $\left(\Omega_{AP}=\tilde{\Omega}_{SA}\right)$. Ratio $T_{AP}/T_{SA}$ (green dotted–dashed line) approaches 5.6 as $\Omega_{AP}$ $\left(\tilde{\Omega}_{SA}\right)$ increases, which indicates the maximum acceleration can be obtained. Difference $T_{AP}-T_{SA}$ is plotted in blue solid line where the maximum shows that the optimal STIRSAP is reached at $\tilde{\Omega}_{SA}/\Omega_0=1.14$.

same high fidelity (99.4%) transfer by sweeping $\Omega_{AP}$ and then compare $T_{AP}$ with the operation time $T_{SA}$ for STIRSAP by sweeping $\tilde{\Omega}_{SA}$. As shown by the green dashed line in Fig. 2d, for the initial Rabi frequency of $\tilde{\Omega}_{SA}=\Omega_0$, which corresponds to a long operation time $T_{SA}$, the auxiliary Rabi frequency $\Omega_a$ is small, resulting in only a slight improvement in $T_{SA}$ (see the time-derivation term in equation (5) in Methods). However, if we slightly increase $\tilde{\Omega}_{SA}$, $\Omega_a$ increases, while the ratio $T_{AP}/T_{SA}$ quickly increases. The ratio is finally stabilized at 5.6, which means that STIRSAP can achieve a speed-up 5.6 times that of STIRAP for a fixed $\Omega_0$. Although the maximum speed-up is achieved when $\tilde{\Omega}_{SA}$ is larger than $2\Omega_0$, an optimal speed-up can be achieved by increasing a moderate factor in $\Omega_0$. We also plot the difference $T_{AP}-T_{SA}$ (in unit of $T_0$) as shown in Fig. 2d (solid blue line), which reaches its maximum when $\tilde{\Omega}_{SA}\approx 1.14\Omega_0$.

**Robustness against imperfection.** We now test the stability of the STIRSAP protocol with respect to control parameter variations. To this end, we experimentally measure and theoretically calculate the transfer efficiency by varying one of the protocol parameters in Hamiltonian (1) (that is, the amplitudes $\tilde{\Omega}_{SA}$ and relative time delay $\Delta\tau$ of the Stokes and pumping pulses, and single-photon detuning $\Delta$) while keeping all other parameters unchanged.

The amplitude of the Raman pulses for each atom in our system is slightly different since there is a space distribution of laser power around $\pm 5\%$ on the atomic cloud. Here we artificially modify the amplitudes of the Raman pulses as $\Omega'_{RR}=\varepsilon\Omega_{RR}$ and $\tilde{\Omega}'_{SA}=\varepsilon\tilde{\Omega}_{SA}$, $\epsilon\in[0.8,1.2]$ (where $RR$ represents

resonant Rabi pulses) to simulate the amplitude variation. Figure 3a shows the experimental data (squares) and theoretical results (lines) of the transfer efficiencies as a function of the deviation $\varepsilon$ for the resonant Raman $\pi$ pulse (green squares and dotted–dashed line), STIRSAP with $T=0.4$ ms (blue squares and dashed line) and STIRSAP with $T=1$ ms (red squares and solid line). As shown in Fig. 3a, the resonant Raman $\pi$ pulse is very sensitive to the amplitude variation of Rabi frequencies, and the maximum transfer efficiency is <90% due to the intensity space distribution of laser fields. Remarkably, the STIRSAP is less sensitive to the change of $\tilde{\Omega}'_{SA}$, since the system adiabatically evolves along the eigenstate of Hamiltonian $H_0$, which depends only on the ratio of the Stokes and pumping fields. The robustness will be improved if we extend $T=0.4-1$ ms, because it will be easier for the system to follow the changes of the ratio of the Stokes and pumping fields.

The transfer efficiencies as a function of the separation time are plotted in Fig. 3b. We first measure the transfer efficiency with fixed pulses shapes versus different separation times $\Delta\tau'$. The pulses of STIRSAP are generated with parameters $\Delta\tau=T/10$ and $T=0.4$ ms. The real separation time $\Delta\tau'$ in our system is achieved by triggering the radio resource with a delay time at a range about $\pm 20\%$ in $\Delta\tau$. We observe the largest 10% reduction in efficiency as shown by the blue squares in Fig. 3b, which accords with the theoretical simulation (blue dashed line). We then measure the transfer efficiency with variable pulse shapes versus different separation times. Here the Raman pulses we use for every separation time are calculated for the STIRSAP according to each specific separation time. Under this condition, the transfer efficiency can be kept to almost 1 as shown by the red curves and squares in Fig. 3b.

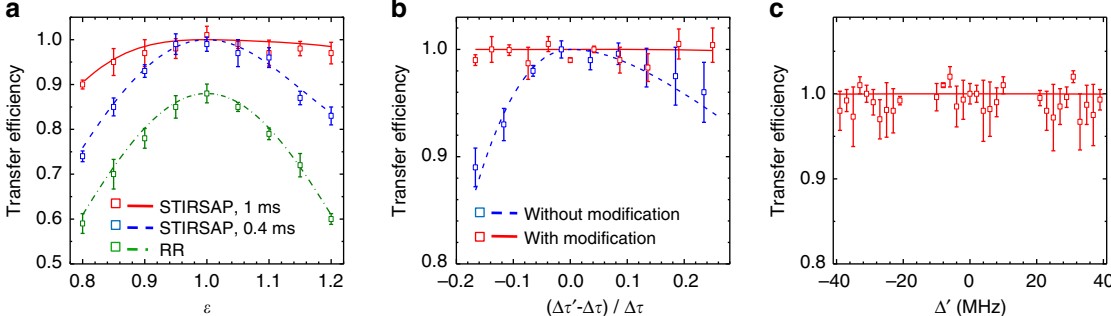

**Figure 3 | Transfer efficiencies of STIRSAP versus three different types of imperfections.** (**a**) Variation in the peak of Rabi frequency characterized with parameter $\varepsilon$. The red solid/blue dashed lines (theoretical results) and the red/blue squares (experimental results) correspond with operation time $T = 1$ 0.4 ms, respectively. Theoretical curve (green dotted–dashed line) and experimental data (green squares) represent the result of resonant Raman (RR) $\pi$ pulse. (**b**) Variation in separation time. Blue squares and dashed lines correspond to variations in $\Delta\tau'$ without pulses shape modification; red squares and solid lines correspond to variations with modification. (**c**) Variation in single-photon detuning. The data points are averaged over five measurements, each with the error bars depicting the s.d.

We further test the sensitivity of the STIRSAP protocol to the variation of the single-photon detuning $\Delta$ in Hamiltonian. The deviation of the detuning is denoted as $\Delta'$ and can be modified in the range of $\pm 40$ MHz in our experiment. The frequency adjustment is implemented by changing the radio frequencies of acousto-optic modulators (AOMs) and the locking points of the pump laser. There are three locking points ($F = 2 \leftrightarrow F' = 2$, $F = 2 \leftrightarrow F' = 3$ and the crossover peak between them) in our set-up, and the radio frequencies of AOMs can be continuously varied $\pm 10$ MHz around each locking point. Although a specific single-photon detuning $\Delta$ is needed in the calculation of the STIRSAP protocol (equation (3)), as shown in Fig. 3c, the transfer efficiency keeps constant as frequency changes, which indicates that STIRSAP will not suffer from the deviation of the detuning $\Delta$, since the variation of $\Delta$ is $<1$ MHz in the experiments.

As discussed above, in the region where the relative imperfection is $<5\%$, STIRSAP with $T = 0.4$ ms can maintain a fidelity higher than 98%, which shows a good robust feature for potential applications in quantum manipulation.

**Double coherent passages and multiple cyclic operation.** So far, we have demonstrated that the STIRSAP protocol is fast, robust and has a high fidelity. As a further proof of its fast and high-fidelity features, we apply STIRSAP pulses at the maximum speed-up point ($T = 0.4$ ms for $\Omega_0$) five times to realize back-and-forth operations in our system. It is noted that the total operation time is limited to 3 ms in our system, mainly due to the expansion of the atomic cloud. For the large single-photon detuning $\Lambda$ system, two coherent passages exit. Thus, the state can be cycled back and forth with the same order of Raman pulses. As shown in Fig. 4a, we first pump all the atoms to one of the ground states ($|1\rangle$) and then repeat the STIRSAP pulse five times. The system will evolve along one eigenstate and then another one. The final population transfer efficiency to the other ground state ($|2\rangle$) is $(95 \pm 4)\%$ averaged over five measured data sets, which indicates an average efficiency of 99(6)%.

More interestingly, the STIRSAP protocol with double coherent passages demonstrated here can also be used to drive the superposition state, which is impossible in ordinary STIRAP with zero detuning. As for an example, we experimentally realize a $\sigma_x$ gate between the initial superposition state $|\psi_0\rangle = \sqrt{0.7}|1\rangle + e^{i\phi_0}\sqrt{0.3}|2\rangle$ and the final state $|\psi_0\rangle = e^{i\phi_0}\sqrt{0.3}|1\rangle + \sqrt{0.7}|2\rangle$ with $\phi_0$ an irrelevant phase. The data driven back and forth for five times are shown in Fig. 4b.

Comparing with the ideal population 0.7 in state $|1\rangle$, the final population measured after five $\sigma_x$ operations is $(68 \pm 4)\%$, which indicates a total transfer efficiency of 96(8)% and an average efficiency of 99(5)%. Note that those multiple cycle operations in Fig. 4a,b cannot be implemented by STIRAP in our system due to the time limit from the expansion of the atomic cloud. The results thus show remarkable advantages of STIRSAP in some quantum systems with short coherent time.

**Discussion**

In summary, we have theoretically proposed and experimentally demonstrated an useful protocol to speed-up conventional 'slow' STIRAP in a large single-photon detuning three-level system through transitionless passage. The STIRSAP demonstrated here is faster than STIRAP and more robust as compared with resonant Raman $\pi$ pulses. Furthermore, the existence of double coherent passages provides a feasible way to control arbitrary quantum states. Fast, high in fidelity and robust against control parameter variations, the STIRSAP protocol is promising for practical applications in quantum control, in quantum information processing and even in chemical interaction control.

**Methods**

**Cold atomic ensemble controlled by Raman lasers.** Our experimental system shown in Fig. 1a is similar to the one described in our previous work[25]. The $^{87}$Rb atoms are trapped by a magneto-optical trap. Two Raman lasers (Stokes and pumping lasers), respectively, couple two ground states ($|1\rangle$, $|2\rangle$) with the excited state ($|3\rangle$). The Raman lasers are set to be two-photon resonance ($\delta = 0$) and large single-photon detuning ($\Delta \sim 2\pi \times 2.5$ GHz from the excited state). The frequency of the Stokes laser is further locked to the pumping laser with a stable beating frequency (bandwidth is $<0.1$ kHz) through optical phase-locked loop technique. The shapes of Raman pulses are controlled by two AOMs (Fig. 1a), which are driven by a radio source (Rigol, DG4162). The radio source has a frequency stability smaller than 2 p.p.m. and a maximum frequency output of 160 MHz.

With a bias field $B_z$ about 0.1 G, two-photon Raman transition between magnetic sublevels of $|F = 1\rangle$ and $|F = 2\rangle$ is split by 140 kHz, which allows us to selectively transfer population between $|F = 1, m_F = 0\rangle$ and $|F = 2, m_{F'} = 0\rangle$. Population is measured with the fluorescence collected by a photodiode. To eliminate the total population fluctuation, the populations of $|F = 1, m_F = 0\rangle$ and $|F = 2, m_{F'} = 0\rangle$ are measured simultaneously in the experiments for normalization.

**Detailed STIRSAP method.** Under the large detuning condition, the three-level $\Lambda$ system reduces to an effective two-level system described by the Hamiltonian (2). According to the theory of shortcut-to-adiabatic passage, the diabatic transition can be eliminated by adding a counter-diabatic term given as $H_{cd}(t) = i\hbar \sum (|\partial_t \lambda_n\rangle\langle\lambda_n| - \langle\lambda_n|\partial_t\lambda_n\rangle|\lambda_n\rangle\langle\lambda_n|)$[16,29], which will lead the system evolution along the eigenstate $|\lambda_n\rangle$ ( $= \{|D\rangle, |B_1\rangle\}$ here ) for any time $T$. For our system, the counter-diabatic term can be realized by adding a microwave field to couple the levels $|1\rangle$ and $|2\rangle$ (refs 29,30). Given this, the counter-diabatic term $H_{cd}$

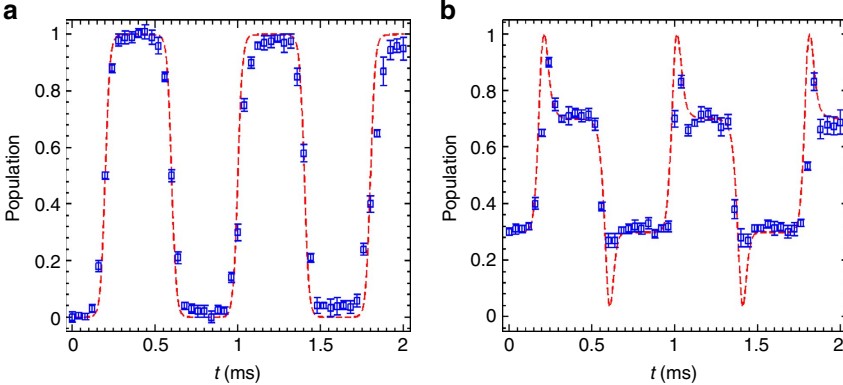

**Figure 4 | Experimental realization of multiple cycle operations.** Populations in state $|2\rangle$ are plotted as a function of time $t$ during five STIRSAP pulses. (**a**) With initial state $|1\rangle$. (**b**) With initial state $|\psi_0\rangle = \sqrt{0.7}|1\rangle + e^{i\phi_0}\sqrt{0.3}|2\rangle$. For both **a** and **b** blue squares are the experimental data, and red dashed lines are theoretical results under ideal conditions. The data points are averaged over five measurements, each with the error bars depicting the s.d.

should be given by

$$H_{cd}(t) = \frac{\hbar}{2}\begin{pmatrix} 0 & \Omega_a(t)e^{i\varphi_a} \\ \Omega_a(t)e^{-i\varphi_a} & 0 \end{pmatrix}, \tag{4}$$

where

$$\Omega_a(t) = \frac{2[\dot{\Omega}_P(t)\Omega_S(t) - \Omega_P(t)\dot{\Omega}_S(t)]}{\Omega_P^2(t) + \Omega_S^2(t)} \tag{5}$$

represents the Rabi frequency of the auxiliary-driving field and its phase $\varphi_a = \varphi_L + \pi/2$. The phase relation requires one to lock the phase between the microwave field and the Raman lasers, which is quite complicated.

To overcome these drawbacks, we develop a much simpler approach to realize the shortcut method to adiabatic passage. We note that $H_{cd}$ can be absorbed into the variation of the original field to form a total Hamiltonian, $H(t) = H_0(t) + H_{cd}(t)$, given by

$$H(t) = -\frac{\hbar}{2}\begin{pmatrix} \Delta_{eff} & \sqrt{\Omega_{eff}^2 + \Omega_a^2}e^{-i\gamma(t)} \\ \sqrt{\Omega_{eff}^2 + \Omega_a^2}e^{i\gamma(t)} & -\Delta_{eff} \end{pmatrix} \tag{6}$$

where $\gamma(t) = \phi(t) + \varphi_L$ with $\phi(t) = \arctan(\Omega_a(t)/\Omega_{eff}(t))$. It implies that the additional microwave field to achieve $H_{cd}$ is not necessary. We may simply modify both the phase and the amplitude of the Raman lasers to effectively add the $H_{cd}$ term and thus realize the shortcut-to-adiabatic passage protocol. Moreover, we further show that the precise control of the time-dependent phase $\gamma(t)$, which is still complicated, can be released. To this end, we apply the unitary transformation[13,17,33]

$$U(t) = \begin{pmatrix} e^{-i\gamma(t)/2} & 0 \\ 0 & e^{i\gamma(t)/2} \end{pmatrix}, \tag{7}$$

which amounts a rotation around the $Z$ axis by $\gamma$ and eliminates the $\sigma_y$ term in the Hamiltonian (6). After the transformation, we obtain an equivalent Hamiltonian with equation (6), $\tilde{H}(t) = U^\dagger H U - i\hbar U^\dagger \dot{U}$, that is,

$$\tilde{H}(t) = -\frac{\hbar}{2}\begin{pmatrix} \tilde{\Delta}_{eff}(t) & \tilde{\Omega}_{eff}(t) \\ \tilde{\Omega}_{eff}(t) & -\tilde{\Delta}_{eff}(t) \end{pmatrix}, \tag{8}$$

where the modified effective detuning $\tilde{\Delta}_{eff}(t) = \Delta_{eff}(t) + \dot{\phi}$ and effective Rabi frequency $\tilde{\Omega}_{eff}(t) = \sqrt{\Omega_{eff}^2(t) + \Omega_a^2(t)}$. In the derivation, $\dot{\varphi}_L = 0$ is used. The wavefunction $|\tilde{\Psi}(t)\rangle$ related to the Hamiltonian $\tilde{H}(t)$ is $|\tilde{\Psi}(t)\rangle = U|\Psi(t)\rangle$, where $|\Psi(t)\rangle$ is the wavefunction related to the Hamiltonian $H(t)$ in equation (6). Since the unitary transformation $U(t)$ is diagonal and the elements are just phase factors, population measured in the basis $\{|1\rangle, |2\rangle\}$ should be the same for both $|\tilde{\Psi}\rangle$ and $|\Psi\rangle$.

An interesting result implied in equation (8) to further simplify the experimental protocol, which will be proven in the next section, is that we can realize shortcut-to-adiabatic passage by replacing $\Omega_S(t)$ and $\Omega_P(t)$ in Hamiltonian (1) with modified Raman pulses $\tilde{\Omega}_S(t), \tilde{\Omega}_P(t)$. By solving the following equations

$$\tilde{\Delta}_{eff}(t) = \frac{\tilde{\Omega}_P^2(t) - \tilde{\Omega}_S^2(t)}{4\Delta}, \\ \tilde{\Omega}_{eff}(t) = \frac{\tilde{\Omega}_P(t)\tilde{\Omega}_S(t)}{2\Delta}, \tag{9}$$

we obtain the results of equation (3). Therefore, we can achieve STIRSAP by replacing the original Raman pulse shapes $\Omega_{S,P}(t)$ with $\tilde{\Omega}_{S,P}(t)$ as described in equation (3).

We should point out that after modifying Raman pulse shapes $\tilde{\Omega}_{S,P}(t)$ the STIRSAP protocol is robust against the control parameter variation but is not necessarily optimal. STIRSAP might be further optimized by using inverse

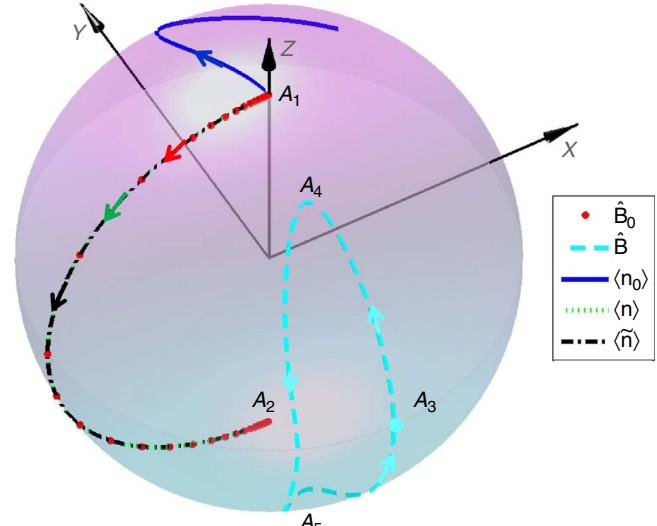

**Figure 5 | Trajectories of the effective magnetic fields and the dynamics of the spin polarizations.** The effective magnetic field $\hat{\mathbf{B}}_0$ (red dot) evolves from the north pole $A_1$ to the south pole $A_2$ along the great circle for the STIRAP protocol. For comparison, the $\hat{\mathbf{B}}$ (cyan dashed line) for STIRSAP started from $A_3$ is also shown. Evolution tracks of the initial state $|1\rangle$ driven by the Hamiltonians $H_0$, $H$ and $\tilde{H}$ are represented by the spin polarizations $\langle \mathbf{n}_0 \rangle$ (blue solid line), $\langle \mathbf{n} \rangle$ (green dotted line) and $\langle \tilde{\mathbf{n}} \rangle$ (black dotted–dashed line), respectively. Since the adiabatic condition is not fully satisfied, $\langle \mathbf{n}_0 \rangle$ does not follow $\hat{\mathbf{B}}_0$. However, both $\langle \mathbf{n} \rangle$ and $\langle \tilde{\mathbf{n}} \rangle$ evolve exactly along the trajectory of $\hat{\mathbf{B}}_0$, as expected by the STIRSAP protocol. The parameters we use to perform numerical simulations are the same as those in Fig. 2a.

engineering[34,35]. Finally, similar STIRSAP protocols can also be implemented with ordinary single-photon resonant STIRAP of the three-level system, which can be reduced to an effective two-level system due to its intrinsic SU(2) symmetry[36].

**Dynamics of the three Hamiltonians.** We here prove that the STIRSAP protocol can be directly achieved by the realization of equation (8). To this end, we compare the dynamics of the three Hamiltonians $H_0(t)$, $H(t)$ and $\tilde{H}(t)$. For any $2 \times 2$ Hamiltonian $H'$, we can relate it with an effective magnetic field $\mathbf{B}'$ by the relation $H' = \frac{1}{2}\sigma \cdot \mathbf{B}'$, that is,

$$\begin{aligned} B_x' &= H_{12}' + H_{21}', \\ B_y' &= i\left(H_{12}' - H_{21}'\right), \\ B_z' &= H_{11}' - H_{22}' \end{aligned} \tag{10}$$

The unit vector of the effective magnetic field is defined as $\hat{\mathbf{B}}' = \mathbf{B}'/|\mathbf{B}'|$. Replaced $H'$ with the Hamiltonian $H_0(t)$ in equation (2) (the Hamiltonian $H(t)$ in equation (6)), we can obtain such effective magnetic field $\hat{\mathbf{B}}_0$ ($\hat{\mathbf{B}}$) for $H_0(t)$ [$H(t)$], and the results are plotted in Fig. 5, where $\Omega_P = \Omega_S = 2\pi \times 5$ MHz, $\Delta = 2\pi \times 2.5$ GHz and $T = 0.4$ ms.

Furthermore, we denote $|\Psi_0(t)\rangle$ as the wavefunction related to the Schrodinger equation $i\hbar\partial_t|\Psi_0(t)\rangle = H_0(t)|\Psi_0(t)\rangle$, and similar denotations for $|\Psi(t)\rangle$ and $|\tilde{\Psi}(t)\rangle$, then the spin polarizations can be defined as

$$
\begin{aligned}
\langle \mathbf{n}_0^{x,y,z}(t)\rangle &= \langle\Psi_0(t)|\sigma_{x,y,z}|\Psi_0(t)\rangle, \\
\langle \mathbf{n}_{x,y,z}(t)\rangle &= \langle\Psi(t)|\sigma_{x,y,z}|\Psi(t)\rangle, \\
\langle \tilde{\mathbf{n}}_{x,y,z}(t)\rangle &= \langle\tilde{\Psi}(t)|\sigma_{x,y,z}|\tilde{\Psi}(t)\rangle
\end{aligned}
\tag{11}
$$

We numerically solve the Schrödinger equations for those Hamiltonians with the initial states given by $|\Psi_0(0)\rangle = |\Psi(0)\rangle = |\tilde{\Psi}(0)\rangle = |1\rangle$ and the initial effective magnetic field $\hat{\mathbf{B}}_0(0)$ is along the $z$ direction. The numerical results of the spin polarizations are plotted in Fig. 5. If the adiabatic condition is fully filled, $\langle\mathbf{n}_0(t)\rangle$ should follow the direction of $\hat{\mathbf{B}}_0(t)$, but as shown in Fig. 5, $\langle\mathbf{n}_0(t)\rangle$ for $T = 0.4$ ms does not overlap $\hat{\mathbf{B}}_0(t)$. However, both $\langle\mathbf{n}(t)\rangle$ and $\langle\tilde{\mathbf{n}}(t)\rangle$ follow along the trajectory of $\hat{\mathbf{B}}_0(t)$. Therefore, rather than following $\mathbf{B}(t)$ or $\tilde{\mathbf{B}}(t)$, both $\langle\mathbf{n}(t)\rangle$ and $\langle\tilde{\mathbf{n}}(t)\rangle$ follow the adiabatic dynamics of the Hamiltonian $H_0(t)$. We thus demonstrate that both $H(t)$ and $\tilde{H}(t)$ can in principle be used to realize STIRSAP protocol, but $\tilde{H}(t)$ is easier to be manipulated in the experiments.

**Data availability.** The data that support the findings of this study are available from the corresponding author on request.

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

## Acknowledgements

We thank Li You for fruitful discussions. This work was supported by the NSF of China (Grants No. 11474107, No. 11474153 and No. 11474193), the NKRDP of China (Grant No. 2016YFA0301800 and No. 2016YFA0302800), the Shuguang Program (Grant No. 14SG35), the SRFDP (Grant No. 2013310811003), the Program for Eastern Scholar, the GDSBPYSTIT (Grant No.2015TQ01X715), the GNSFDYS (Grant No. 2014A030306012), the FOYTHEG (Grant No. Yq2013050), the PRNPGZ (Grant No. 2014010), the PCSIRT (Grant No. IRT1243), NSF of Guangdong province (Grant No. 2016A030310462) and SRFYTSCNU (Grant No. 15KJ15).

## Author contributions

Y.-X.D., Z.-T.L., X.-X.Y., Q.-X.L., W.H. and H.Y. designed and carried out the experiments; Y.-X.D., Z.-T.L., Y.-C.L., X.C. and S.-L.Z. developed the STIRSAP protocol and performed the numerical simulations; Y.-X.D., X.C., H.Y. and S.-L.Z. wrote the paper and all authors discussed the contents; X.C., H.Y. and S.-L.Z. supervised the whole project.

## Additional information

**Competing financial interests:** The authors declare no competing financial interests.

