## [Peer Review File · Nature Communications]

Reviewer #1 (Remarks to the Author):

A) Summary of key results:

The authors achieve a remarkable improvement of STIRAP. By making use of shortcuts-to-adiabaticity (STA) they shorten the times and also increase the stability.

B) There were theoretical proposals to speed up STIRAP but, as the authors point out, they had not been implemented because of several reasons. In this paper the right way to do it has been found, both in theory and experimentally. The new method compares very favorably with respect to the standard approach.

C) The approach is valid, and it works, as demonstrated experimentally. There are some problems with the presentation though. The terminology used is confusing. "Superadiabaticity" is a term that was originally proposed by Berry (not in [16]

but elsewhere!) and then used by various authors in a rather different sense than the one used here. The authors of this paper seem to use it in a loose way (probably following the Pisa group), as a rough synonym of "faster than adiabatic". I strongly discourage the use of this term in that way. "Superadiabaticity" should be kept, according to its original definition, for protocols where a diagonalization of the effective Hamiltonian in the adiabatic basis

is performed to get a superadiabatic basis. This serves to define a new effective Hamiltonian, and actually generates by iteration a family of nested superadiabatic approaches in different orders.

In addition there are some problems with the English that I will comment in point F.

By the way Berry in [16] proposes something that he does not call "transitionless superadiabatic...". The authors seem to mix two very different contributions by Berry.

D) There is an appropriate use of statistics and treatment of uncertainties

E) In conclusion, this is a valuable piece of work, that may have an important impact in different communities and for different applications as sketched in the introduction.

F) Suggested improvements:

- Change the expression "superadiabatic" and the name STIRSAP

- in several places "a microwave"-> "a microwave field" or "pulse" or something similar

- The Methods section is not clear enough and the quality of English is also lower.

An example of the lack of clarity is "the Hamiltonian (6) becomes....". What does "becomes" mean? Is it equal? Is it an effective Hamiltonian? for what transformed wave function? (give more details).

- In a (counterdiabatic) CD-type of approach a CD-term is added to H_0 so that the system following the full Hamiltonian H_0+H_{CD} actually follows the adiabatic dynamics of H_0 . If a unitary transformation is applied to transform H_0+H_{CD} , in general the DYNAMICS IS DIFFERENT (even if the initial and final states are equal by choosing proper boundary conditions for the transformation).

Thus, it is not in principle guaranteed that the new dynamics is adiabatic with respect to H_0 , although this may happen in some cases. This needs clarification -see discussion below(3)-.

- What is "phase temporal"?

- Something wrong with the sentence below (8).

- "can be eliminate" -> "can be eliminated" (in general do a more thorough check of the English)

- as a general rule, do not use "etc" in scientific papers.

G) References are basically OK. Perhaps [13] should be cited the first time that "unitary transformation" is mentioned, as the concept of "unitary transformation" actually used here is rather peculiar and developed in [13].

H) Clarity and context: abstract, Intro, and conclusions are clear and correct except for the point mentioned in C.

Reviewer #2 (Remarks to the Author):

The paper by Du et al. theoretically and experimentally investigates superadiabatic control protocols based on the STIRAP technique. It is found that STIRAP can be made faster and more robust to parameter variations by transforming the pulse shapes of the STIRAP pulses in order to virtually add a counter-diabatic control field. The paper is reasonably clear and well-written, and the results appear to be sound and convincing.

The fact that STIRAP can be improved both in terms of fidelity and robustness is likely to lead to further research and applications, and Nature Communications is an appropriate platform for such a result. Therefore, I can recommend publication of the paper, but would ask the authors to revise their paper taking into account the following points:

- p. 1, LH column: "... the STIRAP ... has several advantages": I'm not sure I understand how STIRAP has an advantage over rapid adiabatic passage because of the possibility to focus the required laser beams on single atoms or ions - surely the same could be done with rapid adiabatic passage, or am I missing something here? The authors should make this point more clearly.

- p. 1, RH column: "... quantum systems that can't be directly coupled...": This statement is a bit vague and misleading. For instance, ground and Rydberg states of an atom can be coupled by a single laser if they have opposite parity, so the general statement the authors are making is not true. Also, the authors probably mean "quantum states" when they write "quantum systems". Could the authors revise this sentence to make it more accurate?

- p. 2, RH column: The authors should give the expression for $H_{cd}(t)$ here, or at least make a reference to the Methods section.

- p. 3, LH column: The transformations of $\Omega_p(t)$ and $\Omega_s(t)$ are very much in the spirit of the superadiabatic transformations of ref. 17, which first introduced them. The authors should acknowledge that when they derive their transformations.

- p. 5, RH column: "The final population is 67.8%..." It is not clear to me how the authors deduce a 96.8% total transfer efficiency from this value; the reasoning behind this should be clearly spelled out.

Apart from the above points, a linguistic revision of the paper is also necessary (examples: p. 1, RH column: "coupling with two partially..." should probably be "... whose internal energy states XX and YY are coupled by ..."; p. 4, LH column: "... with the increase of the Ω_{AP} ..." should be "... with increasing Ω_{AP} ..."; p. 4, RH column: "... while the ratio T_{AP}/T_{SA} quickly ramps up." should be "... quickly increases", etc.

Reviewer #3 (Remarks to the Author):

This paper optimises the shape of laser pulses in a STIRAP experiment. It demonstrates that an optimised shape results in a faster and more robust transfer. This a procedure on which many people have worked before, although the authors appear to be aware of only a small fraction of that work. Some recent key references that are missing in the discussion of the present paper are Phys. Rev. A 85, 033417 (2012) and Phys. Rev. A 86, 023406 (2012). The procedure used in this paper is similar to that employed in the earlier work. However, it lacks some important information, such as what the shape of the "original" Raman pulses chosen as reference is and why this particular shape was chosen. It also contains some misleading or erroneous statements, e.g. "The ultimate goal in quantum control is to prepare a desired state as fast as possible" and "In

most applications, the basic requirement of coherent control is to reach a given target state with high fidelity as fast as possible." : State preparation is certainly not the ultimate goal; gate operations for arbitrary input states are much more important. Similar: "the resulting evolution is robust against control parameter variations when the adiabatic condition is fully satisfied". This is not true in general, only for states commuting with the instantaneous Hamiltonian. The authors try to demonstrate the performance of their scheme for one experiment with a superposition state, but they do not demonstrate the robustness for this state. The statement "Compared with the rapid adiabatic passage, the STIRAP ..." is a meaningless statement; STIRAP itself is an adiabatic passage experiment. Perhaps, the authors have a specific adiabatic passage experiment in mind; this needs to be specified. Some experimental aspects remain unclear, e.g. "the phase difference between the microwave and Raman lasers, which is difficult to lock." Why should this be difficult? They could be generated, e.g., from the same source. They write that they use "ultra-stable radio sources". Specifying the stability in numbers would be much more useful, since "ultra-stable" has very different meanings in different areas. Furthermore, it appears unlikely that the experiment discussed here could be in any way limited by the stability of the "radio source". Overall, I do not recommend publication of this paper.

The lists of the main changes:

1. The term "superadiabatic" in the title and text has been replaced by "shortcut-to-adiabatic".

2. A part titled "Dynamics of the three Hamiltonians" has been newly added in the Methods section, where the validity of our approach has been clearly demonstrated.

3. The typos and grammatical errors mentioned by the referees and found by us have been corrected. Some citations commented by the referees are added.

-----Reply to Reviewer #1 -----

We thank the referee for his/her strong recommendation and detailed suggestions, which are very helpful for improving the manuscript. We have modified the manuscript along the lines suggested by the referee. In the following reply, we address the referee's comments and suggestions point by point:

Comments A and B:

"A) Summary of key results:

The authors achieve a remarkable improvement of STIRAP. By making use of shortcuts-to-adiabaticity (STA) they shorten the times and also increase the stability.

B) There were theoretical proposals to speed up STIRAP but, as the authors point out, they had not been implemented because of several reasons. In this paper the right way to do it has been found, both in theory and experimentally. The new method compares very favorably with respect to the standard approach. "

Reply A and B:

We thank the referee for his/her nice comments.

Comments C:

"C) The approach is valid, and it works, as demonstrated experimentally. There are some problems with the presentation though. The terminology used is confusing. "Superadiabaticity" is a term that was originally proposed by Berry (not in [16] but elsewhere!) and then used by various authors in a rather different sense than the one used here. The authors of this paper seem to use it in a loose way (probably following the Pisa group), as a rough synonym of "faster than adiabatic". I strongly discourage the use of this term in that way. "Superadiabaticity" should be kept, according to its original definition, for protocols where a diagonalization of the effective Hamiltonian in the adiabatic basis is performed to get a superadiabatic basis. This serves to define a new effective Hamiltonian, and actually generates by iteration a family of nested superadiabatic approaches in different orders."

In addition there are some problems with the English that I will comment in point F.

By the way Berry in [16] proposes something that he does not call "transitionless superadiabatic...". The authors seem to mix two very different contributions by Berry."

Reply C:

We thank the referee for his/her detailed suggestion. We agree with the referee's opinion and have deleted the term "superadiabaticity" in the revised manuscript.

Comments D:

"D) There is an appropriate use of statistics and treatment of uncertainties "

Reply D:

The errors have been corrected. $95.2\% \pm 4.1\%$ ($67.8\% \pm 3.8\%$) in the manuscript has been replaced by $(95 \pm 4)\%$ ($68\% \pm 4\%$).

Comment E:

"E) In conclusion, this is a valuable piece of work, that may have an important impact in different communities and for different applications as sketched in the introduction."

Reply E:

We thank the referee for his/her nice comment.

Comments F:

"F) Suggested improvements:

- Change the expression "superadiabatic" and the name STIRSAP*
- in several places "a microwave"-> "a microwave field" or "pulse" or something similar*
- The Methods section is not clear enough and the quality of English is also lower. An example of the lack of clarity is "the Hamiltonian (6) becomes....". What does "becomes" mean? Is it equal? Is it an effective Hamiltonian? for what transformed wave function? (give more details).*
- In a (counterdiabatic) CD-type of approach a CD-term is added to H_0 so that the system following the full Hamiltonian H_0+H_{CD} actually follows the adiabatic dynamics of H_0 . If a unitary transformation is applied to transform H_0+H_{CD} , in general the DYNAMICS IS DIFFERENT (even if the initial and final states are equal by choosing proper boundary conditions for the transformation). Thus, it is not in principle guaranteed that the new dynamics is adiabatic with respect to H_0 , although this may happen in some cases. This needs clarification -see discussion below(3)-.*
- What is "phase temporal"?*

- *Something wrong with the sentence below (8).*
- *"can be eliminate" -> "can be eliminated" (in general do a more thorough check of the English)*
- *as a general rule, do not use "etc" in scientific papers."*

Reply F:

We thank the referee for his/her detailed suggestions. Along the lines suggested by the referee, we have modified the manuscript as follows:

As suggested by the referee, the terminology "superadiabatic" has been deleted. The name STIRSAP has been reserved, but now it is the abbreviation of the phrase "Stimulated Raman Shortcut-to-Adiabatic Passage (STIRSAP)", and thus the title of the manuscript has also been changed.

We agree with the referee's opinion that the dynamics of the three Hamiltonians: H_0 , H_0+H_{CD} and \tilde{H} , are generally different. The validity of the approach adopted here is based on a special condition that the unitary transformation (7) is diagonal and the elements are just phase factors. To clearly show this fact, we newly add a part titled "Dynamics of the three Hamiltonians" in the Methods section, where the trajectories of the spin polarizations for the three Hamiltonians are calculated. The results show that both dynamics of $H(t)$ and $\tilde{H}(t)$ follow the adiabatic dynamics of the Hamiltonian H_0 , as expected.

The minor typos and grammatical errors mentioned by the referee and found by us have been corrected.

Comment G:

"G) References are basically OK. Perhaps [13] should be cited the first time that "unitary transformation" is mentioned, as the concept of "unitary transformation" actually used here is rather peculiar and developed in [13]."

Reply G:

We agree with the referee's opinion and now Ref.[13] has been cited when "unitary transformation" is mentioned.

Comment H:

"H) Clarity and context: abstract, Intro, and conclusions are clear and correct except for the point mentioned in C."

Reply H:

The manuscript has been changed along the lines suggested by the referee.

-----Reply to Reviewer #2 -----

We thank the referee for his/her recommendation and detailed suggestions, which are very helpful for improving the manuscript. We have modified the manuscript along the lines suggested by the referee. In the following reply, we address the referee's comments and suggestions point by point:

Comment 1:

"The paper by Du et al. theoretically and experimentally investigates superadiabatic control protocols based on the STIRAP technique. It is found that STIRAP can be made faster and more robust to parameter variations by transforming the pulse shapes of the STIRAP pulses in order to virtually add a counter-diabatic control field. The paper is reasonably clear and well-written, and the results appear to be sound and convincing.

The fact that STIRAP can be improved both in terms of fidelity and robustness is likely to lead to further research and applications, and Nature Communications is an appropriate platform for such a result. Therefore, I can recommend publication of the paper, but would ask the authors to revise their paper taking into account the following points:"

Reply 1:

We thank the referee for his/her nice comments.

Comment 2:

"- p. 1, LH column: "... the STIRAP ... has several advantages": I'm not sure I understand how STIRAP has an advantage over rapid adiabatic passage because of the possibility to focus the required laser beams on single atoms or ions - surely the same could be done with rapid adiabatic passage, or am I missing something here? The authors should make this point more clearly."

Reply 2:

Thanks for pointing out this mistake. We have deleted "Compared with the rapid adiabatic passage" in this sentence.

Comment 3:

"- p. 1, RH column: "... quantum systems that can't be directly coupled...": This statement is a bit vague and misleading. For instance, ground and Rydberg states of an atom can be coupled by a single laser if they have opposite parity, so the general statement the authors are making is not true. Also, the authors probably mean "quantum states" when they write "quantum systems". Could the authors revise this sentence to make it more accurate?"

Reply 3:

Thanks for pointing out this problem, the related sentence has been revised.

Comment 4:

"- p. 2, RH column: The authors should give the expression for $H_{cd}(t)$ here, or at least make a reference to the Methods section."

Reply 4:

We have added the phrase "defined in the Methods" just after $H_{cd}(t)$.

Comment 5:

"- p. 3, LH column: The transformations of $\Omega_p(t)$ and $\Omega_s(t)$ are very much in the spirit of the superadiabatic transformations of ref. 17, which first introduced them. The authors should acknowledge that when they derive their transformations."

Reply 5:

Thanks for pointing out this issue. Ref.[17] has been cited around Eq.(3) and in the Methods section.

Comment 6 :

"- p. 5, RH column: "The final population is 67.8%..." It is not clear to me how the authors deduce a 96.8% total transfer efficiency from this value; the reasoning behind this should be clearly spelled out. "

Reply 6 :

A reversal σ_x gate is realized here for the initial state $|\psi_0\rangle = \sqrt{0.3}|1\rangle + \sqrt{0.7}|2\rangle$, and thus the final state is $|\psi_0\rangle = \sqrt{0.7}|1\rangle + \sqrt{0.3}|2\rangle$. So the total transfer efficiency here is defined as the ratio between the final population (67.8%) in state $|1\rangle$ measured in the experiment and the ideal transfer efficiency 0.7, that is $0.678/0.7=96.8\%$. We have rewritten this part and clarified this issue in the revised manuscript.

Comment 7:

"Apart from the above points, a linguistic revision of the paper is also necessary (examples: p. 1, RH colum: "coupling with two partially... " should probably be "... whose internal energy states XX and YY are coupled by"; p. 4, LH column: "... with the increase of the Ω_{AP} ..." should be "... with increasing Ω_{AP} ..."; p. 4, RH column: "... while the ratio T_{AP}/T_{SA} quickly ramps up." should be "... quickly increases", etc."

Reply 7:

We have corrected those typos and grammatical errors. Thanks for pointing them out.

-----Reply to Reviewer #3-----

We thank the referee for some of his/her comments. However, we disagree with the referee's criticisms. In the following reply, we address the referee's comments point by point.

Comment 1:

"This paper optimises the shape of laser pulses in a STIRAP experiment. It demonstrates that an optimised shape results in a faster and more robust transfer. This a procedure on which many people have worked before, although the authors appear to be aware of only a small fraction of that work. Some recent key references that are missing in the discussion of the present paper are Phys. Rev. A 85, 033417 (2012) and Phys. Rev. A 86, 023406 (2012). "

Reply 1:

We have read those papers; however, they are just weakly relevant to our work, so we have not cited them in the revised manuscript. The main result in our manuscript is that we theoretically propose and then experimentally demonstrate a shortcut-to-adiabatic scheme to speed up the STIRAP. Therefore, the references we cited are limited in the topic on STIRAP and shortcuts-to-adiabaticity. The references mentioned by the referee are irrelevant to shortcuts-to-adiabaticity. Moreover, as commented by the reviewer A, "References are basically OK" in our manuscript.

Comment 2:

"The procedure used in this paper is similar to that employed in the earlier work. However, it lacks some important information, such as what the shape of the "original" Raman pulses chosen as reference is and why this particular shape was chosen."

Reply 2:

We do not agree with the referee that "The procedure used in this paper is similar to that employed in the earlier work". Especially, our procedure is totally different with those proposed in the references mentioned by the referee. After all, the shortcut-to-adiabatic protocol has not been touched in those papers, see also our reply 1. Furthermore, due to some technical difficulties, the previous proposals of shortcut-to-adiabatic protocol for STIRAP are still hard to be realized, as addressed in our manuscript and our reply 7 below. In conclusion, as commented by both referees #1 and #2, developing a feasible proposal and then demonstrating it should be a remarkable improvement of STIRAP.

On the other hand, the referee missed the information of the "original" Raman pulses. In the last version of our manuscript, the "original" Raman pulses are described in the first sentence in the Results section, that is, the Stokes and pumping pulses are Gaussian-bean pulse shape. Two partially overlapping Gaussian pulses are standard pulse shapes used in STIRAP, as reviewed in Rev.[11] (Rev. Mod. Phys. 70,

1003 (1998). We clarify this issue in the slightly rewritten sentences below Eq.(3).

Comment 3:

"It also contains some misleading or erroneous statements, e.g. "The ultimate goal in quantum control is to prepare a desired state as fast as possible" and "In most applications, the basic requirement of coherent control is to reach a given target state with high fidelity as fast as possible." : State preparation is certainly not the ultimate goal; gate operations for arbitrary input states are much more important."

Reply 3:

In our manuscript, "prepare a desired state" is not limited to "prepare an *initial* state". Similarly, "reach a given target state" is also not limited to "reach an *initial* state". Actually, the double coherent passages demonstrated in our manuscript can be used to realize an arbitrary (single-qubit) gate operation for arbitrary input states, see Ref.[25]. Furthermore, the operation in Fig. 4(b) is actually a σ_x gate for arbitrary input states (see our reply 6 to Reviewer #2). So, our statements mentioned by the referee are reasonable. Anyway, the words "The ultimate goal" there have been changed to "A significantly important goal" in the revised version.

comment 4:

"Similar: "the resulting evolution is robust against control parameter variations when the adiabatic condition is fully satisfied". This is not true in general, only for states commuting with the instantaneous Hamiltonian."

Reply 3:

We think this statement is reasonable. We can't catch the referee's meaning "*only for states commuting with the instantaneous Hamiltonian.*"? Is there any state (Note: not an operator) which can commute with a Hamiltonian?

Comment 5:

"The authors try to demonstrate the performance of their scheme for one experiment with a superposition state, but they do not demonstrate the robustness for this state."

Reply 5:

The robustness in this case has already been demonstrated in this experiment since the operation repeats five times and the total efficiency is larger than 96%. We have mentioned that, due to the expansion of the atomic cloud, the operation should be finished within 3 ms. Five times of the STIRAP with the shortcut-to-adiabaticity can be finished as the total time is $5 \times 0.4 \text{ ms} = 2 \text{ ms}$, while the STIRAP without the shortcut-to-adiabaticity can't be finished in our setup as the evolution time should be at least $5 \times 1.5 \text{ ms} = 7.5 \text{ ms}$.

Comment 6

"The statement "Compared with the rapid adiabatic passage, the STIRAP ..." is a meaningless statement; STIRAP itself is an adiabatic passage experiment. Perhaps, the authors have a specific adiabatic passage experiment in mind; this needs to be specified."

Reply 6:

We thank the referee for pointing out this error. We have deleted the phrase "Compared with the rapid adiabatic passage".

Comment 7 :

"Some experimental aspects remain unclear, e.g. "the phase difference between the microwave and Raman lasers, which is difficult to lock." Why should this be difficult? They could be generated, e.g., from the same source."

Reply 7 :

The referee raised an interesting question why our present protocol is simpler than that using the combination of a microwave field and two Raman laser pulses. As we have mentioned in the Introduction, several protocols have been proposed to speed up the STIRAP by adding an additional microwave field to the usual STIRAP, such as Ref.[29] [X. Chen et al., PRL 105.123003 (2010)]. Actually, in our first try, we used the method proposed in Ref.[29] to do STIRSAP, and do used the same source for the microwave field and for locking the Raman lasers, as suggested by the referee, but still failed because of some technical difficulties. In the STIRAP experiments, we should generate two different pulses shapes for Stokes and pumping pulses, which can't be done in a single acousto-optic modulator (AOM), so two AOMs are used, as shown in Fig. 1a. By using the same source for the microwave field and for locking the Raman lasers, the phases are the same just before the two AOMs and the synchronized point for the microwave field, but different phase noises can be introduced in three separate paths (Stokes laser path, pumping laser path and the path of the microwave field) after the synchronized points.

We may check from the numerical calculation that the transfer efficiency would be high if there are only phase noises from two paths mentioned above, and it is low if there are phase noises from all three paths. With the additional microwave field, the matrix elements representing the coupling between $|1\rangle$ and $|2\rangle$ are not zero. The proposal in Ref.[29] requires a phase coherence between the matrix elements of microwave field and Raman lasers, those phase noises do not fulfill this requirement. From numerical calculation, we find that the transfer efficiency is seriously affected by those phase noises. However, in the absence of the microwave field, the changing rate of the phase noise from only two Raman pulses can be absorbed into the two-photon detuning of the Hamiltonian [see Eq. (8)]. Since the changing rate of the phase noise in our system is typically smaller than the two-photon bandwidth, the transfer efficiency should be robustness against this phase noise. This is the reason why our proposal works, in contrast, the proposal with the additional microwave field

fails even if the same source is used in the experiments.

In principle, this problem can be solved by two ways. One can keep the phase difference between two Raman pulses stable passively by isolating the Raman system after AOMs in a low vibration section. The other way is to realize a feedback setup which can actively return the feedback of the phase noise from two Raman pulses to the microwave field. Both of them are technically complicated. After we failed, we discussed this issue with X. Chen, one of the authors in Ref.[29], and then we developed the present protocol (without the additional microwave field) to avoid this problem, and it works, so we have realized the STIRSAP protocol.

To clarify this issue, we have replaced the sentence “However, the transfer fidelity will depend on the phase difference between the microwave and Raman laser, which is difficult to lock” with the following sentence “ However, the transfer fidelity will depend on the phase differences among the microwave field, the Stokes and pumping laser pulses for the STIRAP, which are difficult to lock”.

Comment 8:

"They write that they use "ultra-stable radio sources". Specifying the stability in numbers would be much more useful, since "ultra-stable" has very different meanings in different areas. Furthermore, it appears unlikely that the experiment discussed here could be in any way limited by the stability of the "radio source"."

Reply 8:

We agree with the referee. We have added the description of the radio sources at the end of the first paragraph of Method section according to the referee's suggestion.

The radio source used here has a frequency stability smaller than 2ppm and a maximum frequency output of 160MHz. The stability of radio source of the AOMs is important in our experiments since it will introduce a shift in two-photon detuning and phase noises if the radio source is unstable.

Comment 9:

"Overall, I do not recommend publication of this paper."

Reply 9:

In summary, we believe that our manuscript is suitable for publication in NC. Furthermore, the manuscript has been recommended by reviewers #1 and #2. We here cite some comments from reviewers #1 and #2:

The Referee #1 commented our manuscript: "The authors achieve a remarkable improvement of STIRAP."; "In conclusion, this is a valuable piece of work, that may have an important impact in different communities and for different applications as sketched in the introduction."

The Referee #2 commented: "The fact that STIRAP can be improved both in terms of fidelity and robustness is likely to lead to further research and applications, and Nature Communications is an appropriate platform for such a result. Therefore, I can recommend publication of the paper,"

Reviewer #1 (Remarks to the Author):

The authors have taken into account my comments and I recommend publication.

Reviewer #2 (Remarks to the Author):

The revised version of the paper takes into account the points I raised in my first review, and also (as far as I can tell) most of those raised by the other two referees. The paper is now more readable and accurate, and I can recommend publication in its present form.

Reviewer #1 (Remarks to the Author):

The authors have taken into account my comments and I recommend publication.

Reply: We thank the referee for his/her recommendation.

Reviewer #2 (Remarks to the Author):

The revised version of the paper takes into account the points I raised in my first review, and also (as far as I can tell) most of those raised by the other two referees. The paper is now more readable and accurate, and I can recommend publication in its present form.

Reply: We thank the referee for his/her recommendation.